# Efficacy of invasive laser acupuncture in treating chronic non-specific low back pain: A randomized controlled trial

Jae-Hong Kim[1,2]☯*, Chang-Su Na[3]☯, Myoung-Rae Cho[1], Gwang-Cheon Park[2], Jeong-Soon Lee[4]

1 Department of Acupuncture and Moxibustion Medicine, College of Korean Medicine, Dongshin University, Naju City, Republic of Korea, 2 Clinical Research Center, Dongshin University Gwangju Korean Medicine Hospital, Gwangju City, Republic of Korea, 3 Department of Acupoint and Meridian, College of Korean Medicine, Dongshin University, Naju City, Republic of Korea, 4 Department of Nursing, Christian College of Nursing, Gwangju City, Republic of Korea

☯ These authors contributed equally to this work.
* nahonga@hanmail.net

**Data Availability Statement:** All relevant data are within the paper and its Supporting Information files.

## Abstract

This study aimed to provide preliminary evidence for the efficacy of invasive laser acupuncture (ILA) for chronic non-specific low back pain (CNLBP). This was a single-center, randomized, patient and assessor-blinded, placebo-controlled, parallel-arm, clinical trial with a 1:1:1 allocation ratio that included a full analysis set. Forty-five participants with CNLBP were randomly assigned to the control group (sham laser), 650 group (650 nm-wavelength ILA), or 830 group (830 nm-wavelength ILA) (n = 15/group). All participants received ILA for 10 min, followed by electroacupuncture for 10 min on the same day. The treatment was performed once per day, twice per week for 4 weeks at bilateral BL23, BL24, BL25, and GB30. The primary outcome was the among-group difference of changes in the visual analog scale (VAS) scores at intervention endpoint (week 4). The secondary outcomes were the among-group difference of changes in VAS at 4 weeks after intervention completion (week 8), those in the Korean version of the Oswestry Disability Index (ODI) and the European Quality of Life Five-Dimension- Five-Level (EQ-5D-5L) at intervention endpoint (week 4) and 4 weeks after intervention completion (week 8). The VAS scores of the 650 group decreased significantly compared with those of the control group (p = 0.047; week 4 vs. week 0). The ODI scores of the 650 group (p = 0.018, week 4 vs. week 0; p = 0.006, week 8 vs. week 0) and 830 group (p = 0.014, week 4 vs. week 0) decreased significantly compared with those of the control group. There was no adverse event related to ILA and no significant difference in changes in vital signs among the three groups. The 650 group showed significant improvements in pain intensity and functional disability. The 830 group showed significant improvements in functional disability. Therefore, ILA therapy at 650 nm and 830 nm wavelengths can be used to treat CNLBP.

**Funding:** This research was supported by a grant of the Korea Health Technology R&D Project through the Korea Health Industry Development Institute (KHIDI), funded by the Ministry of Health & Welfare, Republic of Korea (grant number: HF21C0044). There was no additional external funding received for this study. The funders had no role in study design, data collection and analysis, decision to publish, or preparation of the manuscript.

**Competing interests:** The authors have declared that no competing interests exist.

## Introduction

Chronic non-specific low back pain (CNLBP) is a common health problem; it is considered to be a multifactorial disorder comprising interactions of musculoskeletal pain and psychosocial factors [1,2]. CNLBP is defined as pain and discomfort localized in the lumbosacral region with no specific underlying pathological causes that persists for more than 3 months [3,4]. Functional limitations and consequent disability caused by CNLBP not only affect daily activities and quality of life but also increase health care expenditures [5].

The guidelines for CNLBP recommend treatment comprising the use of non-steroidal anti-inflammatory drugs, antidepressants, exercise therapy, and psychosocial interventions [6,7]. Acupuncture has been used for the treatment of CNLBP mainly by those who practice complementary and alternative medicine [8]. However, the recommendations for acupuncture are inconsistent. Of eight treatment guidelines, only four recommend the use of acupuncture [6].

Laser acupuncture (LA) (the use of a low-level laser [LLL]) to stimulate acupoints) is considered a safe treatment modality because of its painless and noninvasive nature [9]. LA is primarily used to treat arthralgia or musculoskeletal pain; however, some studies have investigated its efficacy for treating CNLBP [10,11]. The efficacy of LLL therapy (LLLT), including LA for CNLBP, has not been established. A Cochrane report [12] concluded that there were insufficient data to draw firm conclusions on the clinical effect of LLLT for low back pain (LBP). There is a need for further methodologically rigorous randomized controlled trials (RCTs) to evaluate the effects of LLLT. Several systematic reviews and meta-analyses [13–15] have suggested that LLLT including LA may have a significant positive effect on reducing pain. However, there is still a lack of high quality evidence supporting its efficacy. Rigorously blinded trials are needed to confirm the efficacy of LLLT for CNLBP and identify the optimal LLLT parameters. Several studies have reported that LA did not show significant effects compared with sham laser for patients with LBP [16–18]. The methods of LLLT used in previous studies were noninvasive and there was no study on invasive laser [12–18].

Noninvasive LA is applied to the skin at acupoints using a laser-emitting device that can be used as an alternative to needles. Invasive LA (ILA) involves the simultaneous application of invasive acupuncture treatment at acupoints and focused laser irradiation using a laser machine connected to an acupuncture needle consisting of an optical fiber-coupled laser diode [19]. Previous studies have shown that ILA has significant effects on neuropathic pain [20,21] and osteoarthritis [22] in rat models. However, clinical evidence obtained from rigorous RCTs regarding the efficacy and safety of ILA for CNLBP is lacking. Therefore, we conducted this pilot study to obtain basic data regarding the efficacy of ILA for CNLBP by comparing the efficacy of ILA at different wavelengths.

## Materials and methods

This study adhered to the standard protocol items of the Recommendations for Interventional Trials (SPIRIT) statements [23] and Consolidated Standards of Reporting Trials (CONSORT) guidelines [24]. The detailed methods of this study have been disclosed in a previous publication [19].

### Study design

This study was a prospective, patient and assessor-blinded, parallel-arm, single-center (Dongshin University Gwangju Korean Medicine Hospital, Republic of Korea) RCT with a 1:1:1 allocation ratio. A total of 45 participants who met the inclusion criteria were randomly allocated to the control group (sham laser; n = 15), 650 group (650 nm-wavelength ILA; n = 15), or 830 group (830 nm-wavelength ILA; n = 15). The participants received ILA for 10 min followed by

electroacupuncture (EA) for 10 min on the same day. The treatment was performed once per day, twice per week, for 4 weeks at bilateral Shenshu (BL23), Qihaishu (BL24), Dachangshu (BL25), and Huantiao (GB30) points.

Outcome measurements were determined at baseline (week 0), 4 weeks after the first intervention (week 4; end of the intervention), and 4 weeks after the completion of the intervention (week 8). The study design is summarized in Table 1.

## Ethical considerations

This study was conducted in accordance with the Declaration of Helsinki. The study protocol (version 1.2) was approved by the Institutional Review Board (IRB) of Dongshin University Gwangju Korean Medicine Hospital (approval no: DSGOH-2019-004; approval date: April 17, 2020) before the trial began. This trial was registered at the Clinical Research Information Service (cris.nih.go.kr; registration number: KCT0004610; registration date: January 7, 2020). The purpose and potential risks of this study were fully explained to all participants, and all participants provided written informed consent before participating in the study.

## Participant recruitment

We recruited participants at Dongshin University Gwangju Korean Medicine Hospital in the Republic of Korea between May 12, 2020 and September 17, 2020. The clinical research

**Table 1. Standard Protocol Items: Recommendations for Interventional Trials (SPIRIT) statement.**

| | STUDY PERIOD | | | | | | | |
|---|---|---|---|---|---|---|---|---|
| | Enrolment | Allocation | Post-allocation | | | | | Close-out |
| **TIMEPOINT** | Screening | | Visit1-2 | Visit3-4 | Visit5-6 | Visit7 | Visit8 | Visit9 |
| | Week | | 1 | 2 | 3 | 4 | 4 | 8 |
| **ENROLMENT** | | | | | | | | |
| Informed consent | X | | | | | | | |
| Sociodemographic profile | X | | | | | | | |
| Medical history | X | | | | | | | |
| Vital signs | X | X | X | X | X | X | X | X |
| Inclusion/exclusion criteria | X | | | | | | | |
| Allocation | | X | | | | | | |
| VAS | X | | | | | | | |
| **INTERVENTIONS** | | | | | | | | |
| ILA(sham, 650 nm, 830 nm) | | | X | X | X | X | X | |
| EA | | | X | X | X | X | X | |
| **ASSESSMENTS** | | | | | | | | |
| Change of medical history | | | X | X | X | X | X | X |
| Safety assessment | | | X | X | X | X | X | X |
| VAS | | | X | | | | X | X |
| Scores for ODI, | | | X | | | | X | X |
| EQ-5D-5L | | | X | | | | X | X |

ILA, invasive laser acupuncture; EA, electroacupuncture; VAS, visual analog scale; ODI, Korean version of the Oswestry Disability Index; EQ-5D-5L, European Quality of Life Five Dimension Five Level Scale.

coordinator (CRC) continuously monitored the medical conditions of the enrolled participants to maximize adherence to the intervention protocols.

## Participation

The inclusion criteria were as follows: age between 19 and 70 years; presence of CNLBP for at least the previous 3 months; score of $\geq$40 points on a 100 mm visual analog scale (VAS) for pain at the time of screening; fluency in Korean language adequate for reliable completion of all study assessments; and voluntary provision of informed consent.

The exclusion criteria were as follows: radicular pain or progressive neurological deficits; diagnosis of a serious spinal pathology (cancer, recent vertebral fracture, spinal infection, or inflammatory spondylitis); presence of a serious chronic disease (cancer; severe cardiovascular, cerebrovascular, liver, or kidney disease; or diabetic neuropathy); history of treatment for alcohol/drug dependency or mental illness (schizophrenia, dementia, or epilepsy) during the 6 months preceding enrollment; LBP not caused by a spinal or soft tissue disease (trauma, ankylosing spondylitis, fibromyalgia, rheumatoid arthritis, or gout); presence of contradictions for LA or EA, such as blood clotting abnormalities (hemophilia), severe skin disease in the lumbar region, presence of metallic devices in the lumbar spine, or presence of electronic medical devices (pacemaker); previous lumbar spinal surgery within 1year or scheduled procedures during the study period; pregnancy or planning to become pregnant; and concurrent participation in other clinical trials.

The dropout criteria were as follows: <75% compliance with the protocol procedures; incidence of serious adverse events (AEs); reluctance to continue participation in the trial; incomplete data that could influence the results; and large protocol error or significant deviation in implementation.

## Randomization and blinding

After acquiring written informed consent and completing baseline measurements, the 45 enrolled participants were assigned serial numbers generated using SPSS version 21 (IBM Corp., Armonk, NY, USA) and randomly allocated to one of the three study groups (n = 15 per group). The serial number codes were inserted in opaque envelopes that were sealed and stored in a double-locked cabinet.

We adopted a patient and assessor-blinded trial procedure. Real laser and sham laser had no differences in appearance, feel, or sound. During the course of this clinical trial, the assessor did not contact any participant at any point of time other than the time of assessment. Hence, participants and assessor remained blinded to the treatment allocation until study completion. A statistician with no conflicts of interest performed the analysis. However, because of the nature of ILA treatment, investigators could not be blinded. The CRC generated the allocation sequence, enrolled the participants, and assigned them to groups.

## Intervention

The participants received treatment once per day, twice per week, for 4 weeks. The treatment was administered by Korean physicians who were licensed to administer treatment and had 6 years of formal university training to practice Korean medicine. ILA and EA treatment were performed using a medical device (Ellise; Wontech Co. Ltd., Daejeon, Republic of Korea) capable of laser irradiation and electrical stimulation. It is composed of a main body consisting of a laser output device and an electrical stimulator, a sterile, stainless steel, disposable acupuncture needle (external diameter, 0.3 mm; inner diameter, 0.15 mm; length, 30 mm) in which an optical fiber is inserted, optical fiber-coupled laser diode (650 nm used the InGaAIP;

830 nm used the GaAIAs) and an electrical stimulus clip. The parameters of the ILA are 20 mW for power, 12 J/point for energy dose, 63.69 W/cm$^2$ for power density, and 38216.56 J/cm$^2$ for energy density. With the participants in the prone position, the needles were vertically inserted in bilateral Shenshu (BL23), Qihaishu (BL24), Dachangshu (BL25), and Huantiao (GB30) points [25,26]. The depths of insertion were between 9 mm and 30 mm, depending on the location of the needle [27]. Noninvasive LA is applied to the skin at acupoints. In contrast, the ILA used in our study was irradiated at the tip of the acupuncture needle under the skin. Manual stimulation was not performed. The participants received ILA (control group, sham laser; 650 group, 650 nm-wavelength laser; 830 group, 830 nm-wavelength laser) for 10 min, followed by EA for 10 min on the same day. The control group underwent the same procedures as the ILA group, but the laser was not turned on. No differences in observations, feelings, or sounds were observed among the three groups during the procedure. Hence, all participants were blinded to the group selection. During the study period, the participants were allowed to continue routine management regimens and existing medications. However, they were not permitted to engage in other treatments to ameliorate CNLBP symptoms.

## Outcome measurements

Primary outcome was the among-group difference of changes in VAS at intervention endpoint (week 4). The secondary outcomes were the among-group difference of changes in VAS at 4 weeks after intervention completion (week 8) and those in the Korean version of the Oswestry Disability Index (ODI) and the European Quality of Life Five-Dimension-Five-Level (EQ-5D-5L) at intervention endpoint (week 4), and 4 weeks after intervention completion (week 8).

The VAS is a 10 cm-long straight line marked at each end with the anchor labels "no pain" and "worst pain imaginable" [28]. It is the most frequently used instrument for measuring the pain intensity of LBP [29].

The ODI is one of the most commonly used scales to assess disability related to LBP [30]. The validated Korean version of the ODI, which excludes items regarding the sexual life of the individual from the original ODI, contains nine questions about daily activities, including inventories of pain intensity, personal care, lifting, walking, sitting, standing, sleeping, social life, and traveling. Each question is rated using a scale of 0 to 5, with a higher score indicating severe pain-related disability [31].

The validated Korean version of the European Quality of Life Five-Dimension (EQ-5D) is a generic instrument used to measure health-related quality of life, including five dimensions pertaining to mobility, self-care, usual daily activities, pain and discomfort, and anxiety and depression [32]. The European Quality of Life Five-Dimension- Five-Level (EQ-5D-5L) is a new version of the EQ-5D scale that includes five levels of severity for each of the five EQ-5D dimensions. Each dimension is scored using a scale of 1 to 5 [33].

## Sample size calculation

Because of the lack of adequate preliminary studies and limited research funds, study period, and recruitment opportunities, we adopted a pilot study design. The appropriate sample size for two-arm or three-arm pilot studies is more than 12 [34,35]. Considering a dropout rate of 20%, we assigned 15 participants to each group (total of 45 participants).

Because our study was a pilot study, the sample size was insufficient to determine the effectiveness of ILA for CNLBP. Our study provides preliminary evidence for the efficacy and safety of ILA for CNLBP, and the results can be used to estimate the appropriate sample size needed for future confirmative RCTs of the efficacy and safety of ILA for CNLBP.

## Statistical analyses

With IRB approval, the statistical analyses were revised during the study. We performed a full analysis set to assess efficacy, and missing values were implemented by the last observation carried forward method. To reduce the risk of bias, a statistician who was not involved in this clinical trial analyzed the final data using SPSS version 20.0 software (SPSS Inc., Chicago, IL, USA) and two-sided significance tests with a 5% significance level. Continuous variables are presented as the mean and standard deviation (SD) or median, first, and third quartiles and categorical variables are presented as the number or percentage.

Baseline characteristics were collected and compared using One-way analysis of variance (ANOVA), the Kruskal-Wallis test or chi-squared ($\chi^2$) test. Within-group differences in all outcome values of the three groups were compared using a repeated-measures ANOVA or Friedman test. Difference in changes in all outcome values among three groups (week 4 vs. week 0 and week 8 vs. week 0) were compared using ANOVA or the Kruskal-Wallis test. Differences in changes in the outcome values between two groups (week 4 vs. week 0 and week 8 vs. week 0; significant changes observed in ANOVA or the Kruskal-Wallis test) were compared using a Scheffe post hoc test or Mann-Whitney U test. For safety evaluation, incidence of adverse events (AEs) among the three groups were compared using chi-squared ($\chi^2$) test and difference in changes in vital signs at 4 weeks after the first intervention (week 4 vs. week 0) were compared among the three groups using ANOVA. Subanalyses and interim analyses were not performed.

## Results

### Participants

We recruited participants between May 11, 2020 and November 5, 2020. During the study period, 388 patients were assessed for eligibility and 343 were excluded. Forty-five patients were included in this study and randomly assigned to the control group (n = 15), 650 group (n = 15), or 830 group (n = 15). Three participants in the control group did not complete the treatment. Two participants in the 830 group did not complete treatment. The data of 45 patients with CNLBP were used for the final analysis (Fig 1).

### Baseline characteristics

The baseline demographic characteristics of the patients in the three groups are presented in Table 2. No significant differences in baseline demographic characteristics or study variables were detected among the three groups (p>0.05) (Table 2).

### Efficacy evaluation

After 4 weeks of intervention, we observed significant improvements in the control group (changes in VAS scores), 650 group (changes in VAS, ODI, and EQ-5D-5L scores), and 830 group (changes in VAS, ODI, and EQ-5D-5L scores) (Table 3).

We observed a significant difference in changes in VAS scores (p = 0.047; week 4 vs. week 0), and ODI scores (p = 0.019, week 4 vs. week 0; p = 0.019, week 8 vs. week 0) among three groups. The VAS scores of the 650 group significantly decreased compared with those of the control group (p = 0.047; week 4 vs. week 0) (Table 4).

Additionally, the ODI scores of the 650 group (p = 0.018, week 4 vs. week 0; p = 0.006, week 8 vs. week 0) and 830 group (p = 0.014, week 4 vs. week 0) significantly decreased compared with those of the control group (Table 5).

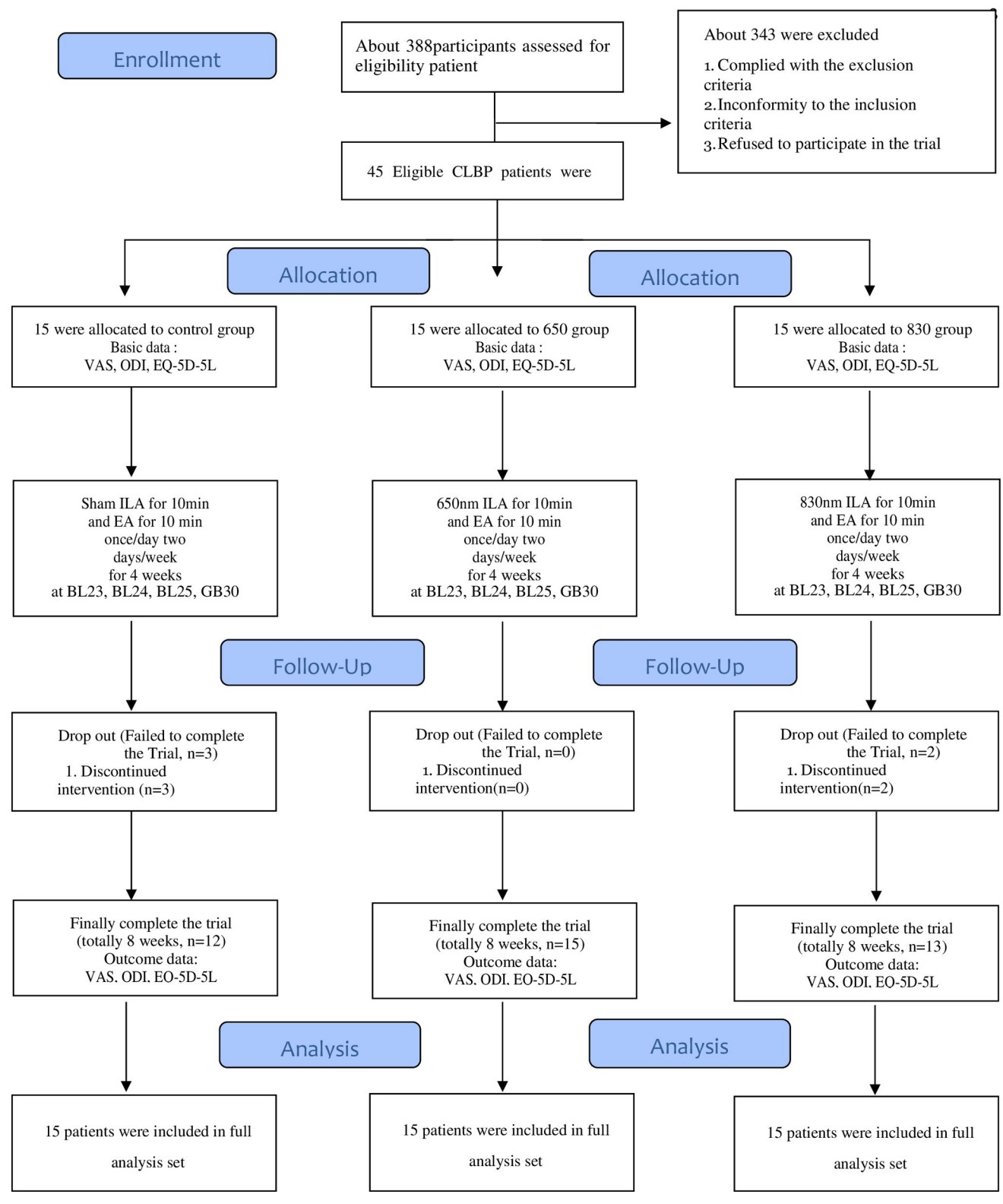

**Fig 1. CONSORT 2010 flow diagram.**

**Table 2. Homogeneity tests for baseline demographic characteristics and study variables for 45 patients with chronic non-specific low back pain.**

| Dependent Variables | control Group (n = 15) | 650 Group (n = 15) | 830 Group (n = 15) | F or χ²(p) |
|---|---|---|---|---|
| | M (SD) or m(Q1, Q3) m (Q1, Q3) or n (%) | M (SD) or m(Q1, Q3) | M (SD) or m(Q1, Q3) | |
| Age (y) | 59.73(6.03) | 57.60(6.85) | 54.40(12.19) | 45.89(0.178) * |
| Sex (Female) | 13 (86.7%) | 9 (60.0%) | 9 (60.0%) | 3.32 (0.190) * |
| Education(years) | 12.67 (4.64) | 12.40 (3.29) | 13.33 (3.70) | 10.99 (0.810) * |
| Height(cm) | 159.75 (5.65) | 162.39 (6.01) | 161.57 (6.20) | 0.77 (0.470) † |
| Weight(kg) | 62.39 (10.86) | 70.43 (9.86) | 65.23 (11.03) | 2.22 (0.121) † |
| VAS | 50.47 (9.38) | 55.07 (17.58) | 49.60 (12.96) | 0.69 (0.508) † † |
| ODI | 9 (7, 11) | 11 (9, 17) | 11 (8, 13) | 3.35 (0.187)¥ |
| EQ-5D-5L | 8 (6, 9) | 9 (8, 11) | 8 (7, 9) | 4.85 (0.088)¥ |

M, Mean; SD, standard deviation; m, median; Q1, first quartiles; Q3, third quartiles.

*, $x^2$-test;

†, One-way ANOVA test;

¥, Kruskal-Wallis test.

## Safety evaluation

For safety evaluation, the incidence of AEs and difference in changes in vital signs at 4 weeks after the first intervention (week 4) were compared among the three groups. AEs that occurred during this study were recorded in a case report form after evaluating their relationship with the intervention. Two AEs related to the intervention occurred in the control group. One was a subcutaneous hematoma and the other was nausea. Both patients recovered without treatment. No serious AEs were observed during this study. There were no significant treatment-induced changes in vital signs, except for some temperature changes in the 650 group. However, the decrease of temperature in the 650 group was within the normal ranges for temperature (Table 6).

**Table 3. Changes in outcome measures after treatment completion in the three groups.**

| Groups | Dependent Variables | Week 0 | Week 4 | Week 8 | χ²(p) |
|---|---|---|---|---|---|
| | | (M±SD) or m(Q1,Q3) | (M±SD) or m(Q1,Q3) | (M±SD) or m(Q1,Q3) | |
| Control group (n = 15) | VAS | 50.47±9.38 | 36.20±19.06 | 34.13±20.65 | **8.28(0.001)** † |
| | ODI | 9 (7,11) | 7 (5,11) | 8 (7,11) | 1.66(0.436) ‡ |
| | EQ-5D-5L | 8 (6,9) | 7 (5,9) | 8 (5,9) | 2.80(0.247) ‡ |
| 650 group (n = 15) | VAS | 55.07 ±17.58 | 23.13 ±11.43 | 26.53 ±23.92 | **15.91(<0.001)** † |
| | ODI | 11 (9,17) | 7 (7,8) | 5 (3,11) | **19.26(<0.001)** ‡ |
| | EQ-5D-5L | 9 (8,11) | 7 (6,8) | 7 (5,9) | **12.50(0.002)** ‡ |
| 830 group (n = 15) | VAS | 49.60 ±12.96 | 28.60 ±14.95 | 25.27 ±15.60 | **21.41(<0.001)** † |
| | ODI | 11 (8,13) | 5 (3,7) | 6 (4,7) | **16.61(<0.001)** ‡ |
| | EQ-5D-5L | 8 (7,9) | 7 (5,7) | 7 (6,7) | **12.84(0.002)** ‡ |

M, Mean; SD, standard deviation; m, median; Q1, first quartiles; Q3, third quartiles.

†, One-way Repeated-measures ANOVA test;

‡, Friedman test.

**Table 4. Difference in changes in VAS scores among three groups (n = 45).**

| Dependent Vatiables | Group | Deference (w4-w0) (M±SD) | F (p) post hoc test (Scheffe) | Deference (w8-w0) (M±SD) | F (p) post hoc test (Scheffe) |
|---|---|---|---|---|---|
| VAS | Control[a] | -14.27±17.17 | 3.29 (0.047) [†] a<b | -16.33±20.12 | 1.37 (0.274) [†] |
| | 650[b] | -31.93±21.94 | | -28.53±24.37 | |
| | 830[c] | -21.00±17.56 | | -24.33±17.17 | |

M, Mean; SD, standard deviation;

[†], One-way ANOVA test.

There were no significant differences in changes in vital signs among the three groups (Table 7).

## Discussion

Although LA has been used to treat CNLBP, the efficacy and optimal treatment method of LA for CNLBP remain controversial because of insufficient evidence. To the best of our knowledge, this is the first RCT to investigate the efficacy of ILA and the effects of different ILA wavelengths on pain intensity, disability related to LBP, and quality of life of patients with CNLBP. Our study results could provide a basis for developing an optimal LA treatment method for CNLBP.

Our study had several main findings. First, the 650 group exhibited a significant improvement in the VAS score compared with the control group at the intervention endpoint. Second, the 650 and 830 groups exhibited significant improvements in the ODI scores compared with the control group at the intervention endpoint. Third, the 650 group exhibited significant improvements in the ODI scores compared with the control group at 4 weeks after intervention completion. Fourth, there was no adverse event related to ILA and no significant difference in changes in vital signs among the three groups. Fifth, we observed significant improvements in the control group (changes in VAS scores), 650 group (changes in VAS, ODI, and EQ-5D-5L scores), and 830 group (changes in VAS, ODI, and EQ-5D-5L scores).

**Table 5. Difference in changes in ODI and EQ-5D-5L scores among three groups (n = 45).**

| Dependent Vatiables | Group | Deference (w4-w0) m(Q1, Q3) | F (p) Z(p) | Deference (w8-w0) m(Q1, Q3) | F (p) Z(p) |
|---|---|---|---|---|---|
| ODI [‡] | Control[a] | -1 (-4, 1) | 7.90 (0.019)[¥] | -1 (-4, 2) | 7.91 (0.019) [¥] |
| | 650[b] | -5 (-10, -1) | | -6 (-8, -2) | |
| | 830[c] | -5 (-7, -1) | **a vs b** **-2.37 (0.018)** [Ŧ] **a vs c** **-2.46 (0.014)** [Ŧ] | -4 (-8, -2) | **a vs b** **-2.73 (0.006)** [Ŧ] a vs c -1.98 (0.047) [Ŧ] |
| EQ-5D-5L [‡] | Control[a] | -1 (-2, 2) | 3.11 (0.211)[¥] | -1 (-2, 2) | 5.78 (0.058)[¥] |
| | 650[b] | -2 (-4, 0) | | -2 (-4, -1) | |
| | 830[c] | -1 (-3, 0) | | -1 (-3, 0) | |

m, median; Q1, first quartiles; Q3, third quartiles;

[¥], Kruskal-Wallis test;

[Ŧ], Mann-Whitney U test P<0.025 (Bonferoni correction).

**Table 6. Changes in vital signs (Week4 vs. Week0) after treatment completion within the three groups.**

| Groups | Dependent Variables | Week 0 (M±SD) | Week 4 (M±SD) | Deference (w4-w0) (M±SD) | t(p)[#] |
|---|---|---|---|---|---|
| Control group (n = 15) | Systoric BP | 124.13±11.96 | 121.47±14.22 | -2.67±9.13 | 1.13(0.277) |
| | Diastoric BP | 82.13±8.42 | 81.80±10.97 | 0.33±9.04 | 0.14(0.889) |
| | Pulse | 73.60±13.02 | 74.80±13.07 | 1.20±7.44 | -0.63(0.625) |
| | Respiration | 20.00±0.00 | 20.00±0.00 | 0.00±0.00 | -0.00(1.00) |
| | Temperature | 36.59±0.30 | 36.52±0.35 | -0.07±0.28 | 1.02(0.326) |
| 650 group (n = 15) | Systoric BP | 125.60±12.44 | 123.53±16.78 | -2.07±11.03 | 0.73(0.480) |
| | Diastoric BP | 79.93±8.64 | 81.00±12.25 | 1.07±9.54 | -0.43(0.671) |
| | Pulse | 75.67±5.61 | 77.13±10.21 | 1.47±8.44 | -0.67(0.512) |
| | Respiration | 20.13±0.52 | 20.00±0.00 | -0.13±0.52 | 1.00(0.334) |
| | Temperature | 36.84±0.21 | 36.64±0.37 | -0.20±0.33 | **2.37(0.033)** |
| 830 group (n = 15) | Systoric BP | 120.73±12.12 | 122.20±16.43 | 1.47±11.54 | -0.49(0.630) |
| | Diastoric BP | 81.40±7.63 | 78.53±9.13 | -2.87±8.15 | 1.36(0.195) |
| | Pulse | 74.67±11.68 | 75.27±11.02 | 0.60±10.68 | -0.22(0.831) |
| | Respiration | 19.87±0.52 | 20.00±0.00 | 0.13±0.52 | -1.00(0.334) |
| | Temperature | 36.66±0.27 | 36.71±0.32 | 0.05±0.24 | -0.87(0.401) |

M, Mean; SD, standard deviation; BP, blood pressure;

[#], paired t- test.

ILA at 650 nm resulted in significant improvements in pain intensity and disability related to pain, and ILA at 830 nm resulted in significant improvements in disability related to pain for patients with CNLBP.

There are several reasons for our results. First, the analgesic effects of LLLT may have influenced the results. LLLT, also known as photobiomodulation, uses red or near-infrared light to stimulate tissue healing and regeneration. Cytochrome c oxidase in the mitochondria and light-/heat-gated ion channels are primary chromophores; both of these lead to the generation of reactive oxygen species that can activate transcription factors, which may act as exercise mimetics [36]. Multiple mechanisms underlie LLLT analgesia. Experimental evidence has suggestedthat laser irradiation induces peripheral neural blockade, suppresses central synaptic

**Table 7. Difference in changes in vital signs (Week4 vs. Week0) among three groups.**

| | Control group | 650 group | 830 group |
|---|---|---|---|
| Systoric BP difference (M±SD) | -2.67±9.13 | -2.07±11.03 | 1.47±11.54 |
| F(p) [†] | | 0.66 (0.520) | |
| Diastoric BP difference (M±SD) | -0.33±9.04 | 1.07±9.54 | -2.87±8.15 |
| F(p) [†] | | 0.75(0.480) | |
| Pulse difference (M±SD) | 1.20±7.44 | 1.47±8.44 | 0.60±10.68 |
| F(p) [†] | | 0.04(0.964) | |
| Respiration difference (M±SD) | 0.00±0.00 | -0.13±0.52 | 0.13±0.52 |
| F(p) [†] | | 1.50 (0.235) | |
| Temperature difference (M±SD) | -0.07±0.28 | -0.20±0.33 | 0.05±0.24 |
| F(p) [†] | | 2.99(0.061) | |

M, Mean; SD, standard deviation;

[†], One-way ANOVA test.

activity, modulates neurotransmitters, reduces muscle spasm and interstitial edema, and exerts anti-inflammatory effects [37]. A recent systematic review and meta-analysis of LLLT (including LA) for CNLBP reported that LLLT has a significant positive effect on reducing pain and improving functional disability and rarely produces severe AEs [15]. Second, the nature of the ILA may have affected the results. Noncollimated light scatters and is reflected at the superficial skin layers, which limits energy penetration through the skin. Because the acupuncture meridians and their acupoints are thought to exist in the myofascial layer of the body, the low energy penetration of light-emitting diode devices theoretically fails to stimulate acupoints [38]. However, ILA can compensate for light scatter and enhance energy penetration because the laser is irradiated at the tip of the acupuncture needle inserted under the skin. During a previous study, 660 nm laser irradiation of the skin did not result in significant pain reduction for patients with LBP [18]. Third, the laser parameters and dosage may have affected the results. The power density and wavelength affect the level of light penetration and scattering. The power density of a laser, defined as the laser energy supplied per area ($W/cm^2$), influences the depth of energy penetration. Higher energy density results in deeper energy transmission [38]. Light wavelengths from 650 to 900 nm penetrate the skin the most. With a well-focused laser beam, red wavelengths (approximately 648 nm) can penetrate 2 to 4 cm beneath the skin surface, and infrared wavelengths (approximately 810 nm) can penetrate up to 6 cm beneath the skin [38].

Our study had some limitations. First, because of limited research funds, study period, and recruitment opportunities, we adopted a pilot study design. The sample size was not sufficient to investigate the efficacy and safety of ILA for CLBP; therefore, this may have led to bias in the results. Second, we did not use the various laser parameters or dosages typically used as treatment of CNLBP. Different wavelengths, power outputs, and energy doses affect the level of light penetration and scattering. There are various laser parameters and dosages for treating CNLBP [10,12]. However, the laser parameters and dosages used during this study are limited. Therefore, further studies investigating effective laser parameters and dosages should be performed. Third, we adopted a patient and assessor-blinded approach because the characteristics of acupuncture application make investigator-blinding impossible.

## Conclusions

ILA at 650 nm led to significant reductions in pain intensity and improvements in pain-related functional disability with CNLBP. ILA at 830 nm led to a significant improvement in pain-related functional disability for patients with CNLBP. Our results suggest that ILA therapy at 650 nm and 830 nm can be used to treat CNLBP. However, because our study was a pilot clinical trial with a small sample size, further rigorously designed clinical studies with a larger sample size are needed to validate these results.

## Supporting information

**S1 Fig. The appearance and components of Ellise.**
(DOCX)

**S2 Fig. Invasive laser acupuncture procedure and electroacupuncture procedure.**
(DOCX)

**S1 File. Data.**
(DOCX)

**S2 File. CONSORT checklist.**
(DOC)

**S3 File. Study protocol.**
(PDF)

## Acknowledgments

The authors express their sincere thanks to their colleagues and the staff at Dongshin University Gwangju Korean Medicine Hospital for their support.

## Author Contributions

**Conceptualization:** Jae-Hong Kim, Chang-Su Na.

**Data curation:** Jae-Hong Kim, Jeong-Soon Lee.

**Formal analysis:** Jeong-Soon Lee.

**Funding acquisition:** Jae-Hong Kim.

**Investigation:** Jae-Hong Kim, Myoung-Rae Cho, Gwang-Cheon Park.

**Methodology:** Jae-Hong Kim, Myoung-Rae Cho.

**Project administration:** Jae-Hong Kim, Chang-Su Na.

**Resources:** Gwang-Cheon Park.

**Software:** Gwang-Cheon Park.

**Supervision:** Jae-Hong Kim, Chang-Su Na.

**Validation:** Jae-Hong Kim.

**Visualization:** Myoung-Rae Cho, Gwang-Cheon Park.

**Writing – original draft:** Jae-Hong Kim, Chang-Su Na.

**Writing – review & editing:** Jae-Hong Kim, Chang-Su Na, Myoung-Rae Cho, Gwang-Cheon Park, Jeong-Soon Lee.

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
