## [Decision Letter · Decision Letter 0]

23 Mar 2022

PONE-D-22-02875Effects of invasive laser acupuncture on chronic non-specific low back pain: A randomized, controlled trialPLOS ONE

Dear Dr. Kim,

Thank you for submitting your manuscript to PLOS ONE. After careful consideration, we feel that it has merit but does not fully meet PLOS ONE’s publication criteria as it currently stands. Therefore, we invite you to submit a revised version of the manuscript that addresses the points raised during the review process.

We look forward to receiving your revised manuscript.

Kind regards,

Yuanyuan Wang, PhD

Academic Editor

PLOS ONE

Journal Requirements:

(Please respond by return e-mail so that we can amend your financial disclosure and competing interests on your behalf."

This research was supported by a grant of the Korea Health Technology R&D Project through the Korea Health Industry Development Institute (KHIDI), funded by the Ministry of Health & Welfare, Republic of Korea (grant number: HF21C0044).The sponsor played no role in the study design, data collection, analysis, interpretation, writing of the report, or the decision to submit the resulting report for publication.)

Additional Editor Comments:

The reviewers have made comments on methodology and introduction. The authors will need to provide more details and make some clarifications.

Reviewers' comments:

Reviewer's Responses to Questions

**Comments to the Author**

1. Is the manuscript technically sound, and do the data support the conclusions?

Reviewer #1: Yes

Reviewer #2: No

Reviewer #3: Yes

2. Has the statistical analysis been performed appropriately and rigorously? 

Reviewer #1: Yes

Reviewer #2: No

Reviewer #3: Yes

3. Have the authors made all data underlying the findings in their manuscript fully available?

Reviewer #1: Yes

Reviewer #2: Yes

Reviewer #3: No

4. Is the manuscript presented in an intelligible fashion and written in standard English?

Reviewer #1: Yes

Reviewer #2: No

Reviewer #3: Yes

5. Review Comments to the Author

Reviewer #1: This is an interesting study. The research project is well prepared and provided.

I have got only a few remarks:

-in Introduction section please describe the power and level of evidence in literature instead of boring and general principles. The analyses of quality of published RCTs so far would be useful. You can describe the PEDro or/and GRADE value of published articles. Please create the table with characteristics of RCTs like in systematic review. You can present Cochrane (if it is possible) reports and metaanalyses/systematic reviews from this area. Generally, the reader would like to read about reasearch status of this area, and what is new in your study. Why it is innovative?

-please enclose a few photos connected with laser app. It would be helpful

Reviewer #2: Thank you for the opportunity to review this interesting topic.

I do have several concerns:

Effects or efficacy? The authors used 'Effects' in the title and then used 'efficacy' in most parts of the paper. They are two different things; please clarify.

However, How was the 'safety' data collected? It was not detailed.

Blinding: patient blind? Single blind? What about the assessor? How did they blind the patients was not clear.

The intervention regime was not clear: All participants received laser acupuncture for 10 min once per day and electroacupuncture for 10 min once per day (twice per week on separate days) for 4 weeks at …; this is very unclear.

ILA and EA were performed with a medical device …. Please provide details of this device.

ILA intervention was not clear; the LA and ILA were used interchangeably. Please provide details of the ILA device.

How did you calculate the energy dose? 12 J/point? 12x8 points? 96 J in total? Where is the recommendation of such dosage?

I am not comfortable with each group having the electroacupuncture for 10 min once per day; the therapeutic effect may be from electroacupuncture, needling, LA, or LLLT since the author declared that the sham group also had significant improvements in the control group changes in VAS scores …

Laser acupuncture and ILA are used interchangeably;

Keywords should have up to 6 keywords and avoid the ones already in the title.

Language editing: there are many misspellings for example, line 53, Comprisinginteractions? Fig.1 words were cut off in half

The authors failed to give due diligence; as such, the report is rather poor in quality.

Reviewer #3: A three arm randomized-control pilot clinical trial aimed to preliminary evaluate the efficacy of two wavelengths of invasive laser acupuncture (ILA) to treat low back pain (CNLBP). The VAS scores of the participants receiving 650-nm wavelength ILA decrease significantly compared to the control group. The ODI scores for both the 650 and 830 wavelength groups decreased significantly compared to the control group.

Minor revisions:

1- Perhaps the abstract should be modified to indicate that the trial aimed to provide preliminary evidence for the efficacy of ILA for CNPBP.

2- Line 190: Clarify what the statisticians were blinded from.

3- If the data is normally distributed provide the mean and standard deviation. However, if the data is not normally distributed, summarize using median, first and third quartiles.

4- Table 3: Define M.

5- Tables: Remove the columns labeled Significant.

6- P-values never equal zero; express small p-values as < 0.001.

7- Table 4: State the type of summary statistics provided in parentheses. Perhaps the label is partially obscured due to formatting of the document.

8- If an interaction effect is significant, provide an interpretation of the results, but do not test main effects because the tests for main effects are uninteresting in light of significant interactions. If interaction effects are non-significant, drop the interaction effects from the model and test the main effects. Determining which results to present when testing interactions is often a multi-step process.

9- Cite the statistical software used for the analysis.

6. PLOS authors have the option to publish the peer review history of their article (what does this mean?). If published, this will include your full peer review and any attached files.

Reviewer #1: **Yes: **Jakub Taradaj

Reviewer #2: No

Reviewer #3: No

---

## [Author Response · Author response to Decision Letter 0]

16 Apr 2022

Reviewer #1: This is an interesting study. The research project is well prepared and provided.

I have got only a few remarks:

-in Introduction section please describe the power and level of evidence in literature instead of boring and general principles. The analyses of quality of published RCTs so far would be useful. You can describe the PEDro or/and GRADE value of published articles. Please create the table with characteristics of RCTs like in systematic review. You can present Cochrane (if it is possible) reports and metaanalyses/systematic reviews from this area. Generally, the reader would like to read about reasearch status of this area, and what is new in your study. Why it is innovative?

Response: Thank you for your valuable comments. In accordance with your suggestion, we have attached a consort checklist as the S2 appendix to indicate reporting quality of our manuscript. We presented the conclusion of Cochrane report and several systematic review/meta-analyses on the efficacy of LLLT for chronic non-specific low back pain to describe the power and level of evidence in literature as follows : “The efficacy of LLL therapy (LLLT), including LA for CNLBP, has not been established. A Cochrane report [12] concluded that there were insufficient data to draw firm conclusions on the clinical effect of LLLT for low back pain (LBP). There is a need for further methodologically rigorous randomized controlled trials (RCTs) to evaluate the effects of LLLT. Several systematic reviews and meta-analyses [13-15] have suggested that LLLT including LA may have a significant positive effect on reducing pain. However, there is still a lack of high quality evidence supporting its efficacy. Rigorously blinded trials are needed to confirm the efficacy of LLLT for CNLBP and identify the optimal LLLT parameters. Several studies have reported that LA did not show significant effects compared with sham laser for patients with LBP [16–18]. The methods of LLLT used in previous studies were noninvasive and there was no study on invasive laser [12-18]. on pages 3-4, lines 65-76.

-please enclose a few photos connected with laser app. It would be helpful

Response: Thank you for your comments. Per your suggestion, we have attached photos of the appearance and components of Ellise, invasive laser acupuncture (ILA) procedure, and electroacupuncture procedure as S1 Fig and S2 Fig. 

Reviewer #2: Thank you for the opportunity to review this interesting topic.

I do have several concerns:

Effects or efficacy? The authors used 'Effects' in the title and then used 'efficacy' in most parts of the paper. They are two different things; please clarify.

Response: We apologize for the confusion. We have revised the title as follows : “Efficacy of invasive laser acupuncture in treating chronic non-specific low back pain: A randomized, controlled trial” on pages 1, lines 1-2.

However, How was the 'safety' data collected? It was not detailed.

Response: We apologize for not being detailed in our description of safety evaluation. We inserted the pertinent sentences in the statistical analyses and safety evaluation subsection as follows : “For safety evaluation, incidence of adverse events (AEs) among the three groups were compared using chi-squared (χ2) test and difference in changes in vital signs at 4 weeks after the first intervention (week 0 vs. week 4) were compared among the three groups using ANOVA.” on page 12, lines 238-241.

“For safety evaluation, the incidence of AEs and difference in changes in vital signs at 4 weeks after the first intervention (week 4) were compared among the three groups. AEs that occurred during this study were recorded in a case report form after evaluating their relationship with the intervention. Two AEs related to the intervention occurred in the control group. One was a subcutaneous hematoma and the other was nausea. Both patients recovered without treatment. No serious AEs were observed during this study. There were no significant treatment-induced changes in vital signs except for some temperature changes in the 650 group. However, the decrease of temperature in the 650 group was within the normal ranges for temperature (Table 6). There were no significant differences in changes in vital signs among the three groups (Table 7)” on page 16-18, line 291-306.

We have inserted the changes in vital signs within the three groups and difference in changes in vital sign among the three groups as Table 6 and Table 7. 

Blinding: patient blind? Single blind? What about the assessor? How did they blind the patients was not clear.

Response: Thank you for your query. We apologize for the confusion. We adopted a patient and assessor-blinded trial procedure. We revised Randomization and Blinding subsection as follows: “We adopted a patient and assessor-blinded trial procedure. Real laser and sham laser had no differences in appearance, feel, or sound. During the course of this clinical trial, the assessor did not contact any participant at any point of time other than the time of assessment. Hence, participants and assessor remained blinded to the treatment allocation until study completion. A statistician with no conflicts of interest performed the analysis. However, because of the nature of ILA treatment, investigators could not be blinded.”

on page 8, line 156-161.

We described the patient blinding method in the intervention subsection as follows : “The control group underwent the same procedures as the ILA group, but the laser was not turned on. No differences in observations, feelings, or sounds were observed among the three groups during the procedure. Hence, all participants were blinded to the group selection.” on page 9, line 182-185.

The intervention regime was not clear: All participants received laser acupuncture for 10 min once per day and electroacupuncture for 10 min once per day (twice per week on separate days) for 4 weeks at …; this is very unclear.

Response: We apologize for not clearly describing the intervention regime. We revised this sentence for more clarity as follows : “ All participants received laser acupuncture for 10 min, followed by electroacupuncture for 10 min on the same day. The treatment was performed once per day, twice per week for 4 weeks at bilateral BL23, BL24, BL25, and GB30.” on page 2, line 31-34.

ILA and EA were performed with a medical device …. Please provide details of this device.

Response: Thank you for your comments. To provide details of a medical device, we attached photos of the appearance and components of Ellise, invasive laser acupuncture(ILA) procedure and electroacupuncture procedure as S1 Fig and S2 Fig, and we have revised the intervention subsection to provide details of a medical device as follows : “ILA and EA treatment were performed using a medical device ( Ellise; Wontech Co. Ltd., Daejeon, Republic of Korea) capable of laser irradiation and electrical stimulation. It is composed of a main body consisting of a laser output device and an electrical stimulator, a sterile, stainless steel, disposable acupuncture needle (external diameter, 0.3 mm; inner diameter, 0.15 mm; length, 30 mm) in which an optical fiber is inserted, optical fiber-coupled laser diode (650 nm used the InGaAIP; 830 nm used the GaAIAs) and an electrical stimulus clip. The parameters of the ILA are 20 mW for power, 12 J/point for energy dose, 63.69 W/cm2 for power density, and 38216.56 J/cm2 for energy density. on page 9, lines 167-175.

ILA intervention was not clear; the LA and ILA were used interchangeably. Please provide details of the ILA device.

Response: Thank you for your comments. LA used in RCT for CNLBP is noninvasive. LA is applied to the skin at acupoints using a laser-emitting device that can be used as an alternative to acupuncture needles [1-7]. The laser beams irradiated to the skin are scattered and reflected at the superficial skin layers, which limits energy penetration through the skin. In contrast, ILA used in our study is irradiated at the tip of the acupuncture needle under the skin. Thus, ILA can compensate for light scatter and enhance energy penetration. 

We inserted the pertinent sentences in the intervention subsection as follows : “Noninvasive LA is applied to the skin at acupoints. In contrast, the ILA used in our study was irradiated at the tip of the acupuncture needle under the skin.” on page 9, line 178-180.

For a detailed description of the ILA device, please see the answer to the above questions.

How did you calculate the energy dose? 12 J/point? 12x8 points? 96 J in total? Where is the recommendation of such dosage?

Response: Thank you for your query. The following physical formula was used to calculate the dose [2]: Energy dose (J) = Watts (W) ⅹseconds (secs); Power density (W/cm2) = Watts/area of acupuncture needle tip (cm2); Energy density (J/ cm2) = Watts (W) ⅹseconds (secs)/ area of acupuncture needle tip (cm2)

The parameters of the ILA used in our study are as follows: 

- Energy dose : 12(J) = 0.02 (W) ⅹ 600 (secs) 

- Power density : 63.69 W/cm2 = 0.02 (W) / 3.14 ⅹ 0.01 ⅹ 0.01(cm2)

- Energy density : 38216.56 J/cm2 = 0.02 (W) ⅹ 600 (secs)/ 3.14 ⅹ 0.01 ⅹ 0.01(cm2)

The energy dose used in our study was 12 J/point and 96 J (12 ⅹ 8 points) in total. 

There is no energy dose of recommended LLLT used in CNLBP treatment. The energy dose used in RCTs that investigated the efficacy of LLLT including LA for CNLBP varied to 0.2 – 1873.8 J/point [4]. A recent systematic review and meta-analysis suggested that the energy dose of LLLT for CNLBP was recommended to be 4 J or more per point [8].

I am not comfortable with each group having the electroacupuncture for 10 min once per day; the therapeutic effect may be from electroacupuncture, needling, LA, or LLLT since the author declared that the sham group also had significant improvements in the control group changes in VAS scores …

Response: Thank you for your comments. We understand your concern that electroacupuncture treatment would have affected the results of our study. When we designed this study, we thought that all participants in our study should receive the minimal treatment for CNLBP. Thus, we adopted electroacupuncture as the basic treatment. Exercise, massage, hot fomentation, and soft cupping were performed in both groups as basic treatment in the previous RCTs that investigated the efficacy of LLLT for CNLBP [1-4]. 

Laser acupuncture and ILA are used interchangeably;

Response: Thank you for your comments. LA is noninvasive. LA is applied to the skin at acupoints using a laser-emitting device that can be used as an alternative to acupuncture needles. In contrast, ILA used in our study is irradiated at the tip of the acupuncture needle under skin. Ellise, a medical device used in our study, is only able to perform ILA using a sterile, stainless steel, disposable acupuncture needle in which an optical fiber is inserted. The acupuncture needles were vertically inserted in acupoints and then laser was irradiated at the tip of the acupuncture needle under skin. We have attached photos of the invasive laser acupuncture (ILA) procedure as S2 Fig to describe the ILA procedure in detail.

Keywords should have up to 6 keywords and avoid the ones already in the title.

Response: Thank you for your comments. In accordance with your advice, we have revised the keywords as follows : “low back pain, 650 nm low level laser, 830 nm low level laser, pain intensity, visual analog scale, clinical trial”

Language editing: there are many misspellings for example, line 53, Comprisinginteractions? Fig.1 words were cut off in half

The authors failed to give due diligence; as such, the report is rather poor in quality.

Response: We appreciate your pertinent observations. Based on your observations, we have carefully rechecked the revised manuscript for further grammar and spelling errors. We have had our manuscript re-edited by a professional English language editing service to ensure that the text is free from any spelling and grammar errors. The edit was performed by editors at Editage, a division of Cactus Communications.

Reviewer #3: A three arm randomized-control pilot clinical trial aimed to preliminary evaluate the efficacy of two wavelengths of invasive laser acupuncture (ILA) to treat low back pain (CNLBP). The VAS scores of the participants receiving 650-nm wavelength ILA decrease significantly compared to the control group. The ODI scores for both the 650 and 830 wavelength groups decreased significantly compared to the control group.

Minor revisions:

1- Perhaps the abstract should be modified to indicate that the trial aimed to provide preliminary evidence for the efficacy of ILA for CNLBP.

Response: Thank you for your comments. In accordance with your suggestion, we inserted the following sentence into the abstract section: “ This study aimed to provide preliminary evidence for the efficacy of invasive laser acupuncture (ILA) for chronic non-specific low back pain (CNLBP).” on page 2, line 26-27.

2- Line 190: Clarify what the statisticians were blinded from.

Response: We apologize for the confusion. We revised “Analyses were performed by blinded biostatisticians” to “To reduce risk of bias, a statistician who was not involved in this clinical trial analyzed the final data.” on page 11, line 226-227.

3- If the data is normally distributed provide the mean and standard deviation. However, if the data is not normally distributed, summarize using median, first and third quartiles.

Response: Thank you for pointing this out. VAS data is normally distributed, while ODI and EQ-5D-5L data are not normally distributed. Therefore, based on your advice, we provided VAS and vital signs data as mean and standard deviation, and ODI and EQ-5D-5L data as median, first, and third quartiles in tables 2, 3, 4, 5, 6, and 7. 

4- Table 3: Define M.

Response: We apologize for the confusion. We have provided mean as M and median as m to prevent confusion and we have defined M and m below the table 2, 3, 4, 5, 6, and 7. 

5- Tables: Remove the columns labeled Significant.

Response: Thank you for your comments. We revised Table 4 that analyzed interaction between time and group to Table 4, which compared changes in the VAS scores among three groups.

6- P-values never equal zero; express small p-values as < 0.001.

Response: Thank you for your comments. According to your advice, we expressed small p-values as < 0.001 in Table 3.

7- Table 4: State the type of summary statistics provided in parentheses. Perhaps the label is partially obscured due to formatting of the document.

Response: Thank you for your comments. We revised Table 4, which analyzed interaction between time and group to Table 4, which compared changes in the VAS scores among three groups.

8- If an interaction effect is significant, provide an interpretation of the results, but do not test main effects because the tests for main effects are uninteresting in light of significant interactions. If interaction effects are non-significant, drop the interaction effects from the model and test the main effects. Determining which results to present when testing interactions is often a multi-step process.

Response: We appreciate your valuable suggestions. The purpose of our study was to investigate the efficacy of ILA on pain reduction in CNLBP. Therefore, the primary outcome was the among-group difference of changes in VAS at 4 weeks after the first intervention (week 4). The interaction between time and group of VAS was not significant. Based on your suggestions, we have revised Table 4, which analyzed interaction between time and group to Table 4, which compared changes in VAS scores among the three groups. 

9- Cite the statistical software used for the analysis.

Response: Thank you for your comments. We have cited the statistical software used for the analyses in the statistical analyses subsection as follows : “To reduce risk of bias, a statistician who was not involved in this clinical trial analyzed the final data using SPSS version 20.0 software (SPSS Inc., Chicago, IL, USA) and two-sided significance tests with a 5% significance level.” on page 11, line 226-228.

Reference

1. Yousefi-Nooraie R, Schonstein E, Heidari K, Rashidian A, Pennick V, Akbari-Kamrani M, et al. Low level laser therapy for nonspecific low-back pain. Cochrane Database Syst Rev. 2008;2: CD005107. doi: 10.1002/14651858.CD005107.pub4. 

2. Glazov G, Yelland M, Emery J. Low-level laser therapy for chronic non-specific low back pain: a meta-analysis of randomised controlled trials. Acupunct Med. 2016;34: 328-341. doi: 10.1136/acupmed-2015-011036. 

3. Huang Z, Ma J, Chen J, Shen B, Pei F, Kraus VB. The effectiveness of low-level laser therapy for nonspecific chronic low back pain: a systematic review and meta-analysis. Arthritis Res Ther. 2015;17: 360. doi: 10.1186/s13075-015-0882-0.

4. Yeum H, Hong Y, Nam D. Low-level laser therapy including laser acupuncture for non-specific chronic low back pain: systematic review and meta-analysis. J Acupunct Res. 2021;38: 8-19. doi: 10.13045/jar.2020.00283. 

5. Glazov G, Yelland M, Emery J. Low-dose laser acupuncture for non-specific chronic low back pain: a double-blind randomised controlled trial. Acupunct Med. 2014;32: 116-123. doi: 10.1136/acupmed-2013-010456. 

6. Glazov G, Schattner P, Lopez D, Shandley K. Laser acupuncture for chronic non-specific low back pain: a controlled clinical trial. Acupunct Med. 2009;27: 94-100. doi: 10.1136/aim.2009.000521.

7. Shin JY, Ku B, Kim JU, Lee YJ, Kang JH, Heo H, et al. Short-term effect of laser acupuncture on lower back pain: a randomized, placebo-controlled, double-blind trial. Evid Based Complement Alternat Med. 2015;2015: 808425. doi: 10.1155/2015/808425.

8. Yeum H, Hong Y, Nam D. Effective low-level laser therapy including laser acupuncture treatment conditions for non-specific chronic low back pain: systematic review and meta-analysis. J Acupunct Res. 2021;38: 85-95. doi: 10.13045/jar.2020.00311.

---

## [Decision Letter · Decision Letter 1]

12 May 2022

PONE-D-22-02875R1Efficacy of invasive laser acupuncture in treating chronic non-specific low back pain: A randomized, controlled trialPLOS ONE

Dear Dr. Kim,

Thank you for submitting your manuscript to PLOS ONE. After careful consideration, we feel that it has merit but does not fully meet PLOS ONE’s publication criteria as it currently stands. Therefore, we invite you to submit a revised version of the manuscript that addresses the points raised during the review process.

We look forward to receiving your revised manuscript.

Kind regards,

Yuanyuan Wang, PhD

Academic Editor

PLOS ONE

Journal Requirements:

Additional Editor Comments (if provided):

The authors have addressed all the reviewers' comments. Some minor edits are needed.

Reviewers' comments:

Reviewer's Responses to Questions

**Comments to the Author**

1. If the authors have adequately addressed your comments raised in a previous round of review and you feel that this manuscript is now acceptable for publication, you may indicate that here to bypass the “Comments to the Author” section, enter your conflict of interest statement in the “Confidential to Editor” section, and submit your "Accept" recommendation.

Reviewer #2: All comments have been addressed

2. Is the manuscript technically sound, and do the data support the conclusions?

Reviewer #2: Yes

3. Has the statistical analysis been performed appropriately and rigorously? 

Reviewer #2: Yes

4. Have the authors made all data underlying the findings in their manuscript fully available?

Reviewer #2: Yes

5. Is the manuscript presented in an intelligible fashion and written in standard English?

Reviewer #2: Yes

6. Review Comments to the Author

Reviewer #2: The paper reads much clearer after the revision.

A few minor changes are required:

page 11, line 220: the first word 'efficacy' should be 'effectiveness':

"Because our study was a pilot study, the sample size was insufficient to determine the efficacy of ILA for CNLBP.".

Table 3 needs editing; some words are cut in half. Same as Table 7.

Table 6, there should be a gap between 'group' and (n=15).

In figure 1, the first letter should be F, not f.

7. PLOS authors have the option to publish the peer review history of their article (what does this mean?). If published, this will include your full peer review and any attached files.

Reviewer #2: No

---

## [Author Response · Author response to Decision Letter 1]

14 May 2022

We are very grateful for the constructive comments and the opportunity to revise our manuscript. 

Point-by-point responses to the reviewers’ comments are provided below. The corresponding changes have been highlighted in the revised manuscript with track changes.

6. Review Comments to the Author

Reviewer #2: The paper reads much clearer after the revision.

A few minor changes are required:

page 11, line 220: the first word 'efficacy' should be 'effectiveness':

"Because our study was a pilot study, the sample size was insufficient to determine the efficacy of ILA for CNLBP.".

Response. Thank you for your comments. In accordance with your suggestion, We revised ‘efficacy’ to ‘effectiveness’

Table 3 needs editing; some words are cut in half. Same as Table 7.

Response. Thank you for your comments. In accordance with your suggestion,we edited Table 3 and 7. 

Table 6, there should be a gap between 'group' and (n=15).

Response. Thank you for your comments. In accordance with your suggestion,we edited Table 6. 

In figure 1, the first letter should be F, not f.

Response. Thank you for your comments. In accordance with your suggestion,we revised the figure 1.

---

## [Editor Report · Decision Letter 2]

18 May 2022

Efficacy of invasive laser acupuncture in treating chronic non-specific low back pain: A randomized, controlled trial

PONE-D-22-02875R2

Dear Dr. Kim,

We’re pleased to inform you that your manuscript has been judged scientifically suitable for publication and will be formally accepted for publication once it meets all outstanding technical requirements.

Kind regards,

Yuanyuan Wang, PhD

Academic Editor

PLOS ONE

Additional Editor Comments (optional):

The authors have addressed all the reviewers' comments.
---

## [Editor Report · Acceptance letter]

20 May 2022

PONE-D-22-02875R2 

Efficacy of invasive laser acupuncture in treating chronic non-specific low back pain: A randomized controlled trial

Dear Dr. Kim:

I'm pleased to inform you that your manuscript has been deemed suitable for publication in PLOS ONE. Congratulations! Your manuscript is now with our production department. 

Kind regards, 

on behalf of

Dr. Yuanyuan Wang 

Academic Editor

PLOS ONE